# Habitat Distribution Pattern of Rare and Endangered Plant *Magnolia wufengensis* in China under Climate Change

Xiaodeng Shi [1,2,3,†], Qun Yin [2,3,†], Ziyang Sang [4], Zhonglong Zhu [2,3], Zhongkui Jia [2,3,*] and Luyi Ma [2,3,*]

1    Zhejiang Academy of Forestry, Hangzhou 310023, China
2    Magnolia Wufengensis Research Center, Beijing Forestry University, Beijing 100083, China
3    Key Laboratory for Silviculture and Conservation of the Ministry of Education, College of Forestry, Beijing Forestry University, Beijing 100083, China
4    Wufeng Magnolia Technology Development Co., Ltd., Yichang 443400, China
*    Correspondence: jiazk@163.com (Z.J.); maluyi@bjfu.edu.cn (L.M.)
†    These authors contributed equally to this work.

**Abstract:** *Magnolia wufengensis* is a newly discovered rare and endangered species endemic to China. The primary objective of this study is to find the most suitable species distribution models (SDMs) by comparing the different SDMs to predict their habitat distribution for protection and introduction in China under climate change. SDMs are important tools for studying species distribution patterns under climate change, and different SDMs have different simulation effects. Thus, to identify the potential habitat for *M. wufengensis* currently and in the 2050s (2041–2060) and 2070s (2061–2080) under different climate change scenarios (representative concentration pathways RCP2.6, RCP4.5, RCP6.0, and RCP8.5) in China, four SDMs, Maxent, GARP, Bioclim, and Domain, were first used to compare the predicted habitat and explore the dominant environmental factors. The four SDMs predicted that the potential habitats were mainly south of 40° N and east of 97° E in China, with a high distribution potential under current climate conditions. The area under the receiver operating characteristic (ROC) curve (AUC) (0.9479 ± 0.0080) was the highest, and the Kappa value (0.8113 ± 0.0228) of the consistency test and its performance in predicting the potential suitable habitat were the best in the Maxent model. The minimum temperature of the coldest month (−13.36–9.84 °C), mean temperature of the coldest quarter (−6.06–12.66 °C), annual mean temperature (≥4.49 °C), and elevation (0–2803.93 m), were the dominant factors. In the current climate scenario, areas of 46.60 × 10$^4$ km$^2$ (4.85%), 122.82 × 10$^4$ km$^2$ (12.79%), and 96.36 × 10$^4$ km$^2$ (10.03%), which were mainly in central and southeastern China, were predicted to be potential suitable habitats of high, moderate, and low suitability, respectively. The predicted suitable habitats will significantly change by the 2050s (2040–2060) and 2070s (2060–2080), suggesting that *M. wufengensis* will increase in high-elevation areas and shift northeast with future climate change. The comparison of current and future suitable habitats revealed declines of approximately 4.53%–29.98% in highly suitable habitats and increases of approximately 6.45%–27.09% and 0.77%–21.86% in moderately and lowly suitable habitats, respectively. In summary, these results provide a theoretical basis for the response to climate change, protection, precise introduction, cultivation, and rational site selection of *M. wufengensis* in the future.

**Keywords:** climate change; environmental factors; introduction; *Magnolia wufengensis*; species distribution models; suitable habitats





## 1. Introduction

According to the Sixth Assessment Report (AR6) of the Intergovernmental Panel on Climate Change (IPCC), the global surface temperature from 2011 to 2020 is 1.1 °C higher than that from 1850 to 1900, and the global temperature rise may reach 1.5 °C, or face the risk of temporarily breaching 1.5 °C in the near term [1]. The escalating global surface temperature presents a profound and imminent threat to the survival of

plants [2], which will lead to the reduction of the suitable habitat area for some plants, habitat fragmentation, and even the acceleration of the loss of global biodiversity [3–5]. Rare and endangered plants (REPs) constitute a vital component of biodiversity, playing a pivotal role in maintaining ecosystem health [6,7]. REPs are highly susceptible to environmental changes due to their special physiological characteristics [8,9], such as narrow distribution ranges, high environmental requirements, weak natural regeneration, and few genetic resources [10,11]. According to recent research, approximately one-third of the world's tree species face the risk of extinction primarily due to human activity and climate change-induced habitat loss [12]. Over a hundred tree species have already vanished from the wild, and unless immediate and decisive action is taken, many more are on the brink of extinction [13]. The protection of rare and endangered plants is the premise and basis for the protection of biodiversity and should be one of the most urgent tasks at present.

*Magnolia wufengensis* is a rare and endangered species of the Magnoliaceae family that was published in 2006 [14,15]; it is a group-building tree in forest ecosystems. *M. wufengensis* is a lofty, deciduous arbor species with abundant variations in flower shape, flower color, and petal number. The shapes of its tepals are ovoid, obovate, narrowly obovate, and long lanceolate. The tepal color changes from white and pale red to red and purple–red; the number of tepals varies from 9 to 46 [16]. Thus, *M. wufengensis* has great aesthetic value, can be used for urban greening and mountain afforestation, and has great potential for promotion and utilization.

However, the natural distribution range of *M. wufengensis* is very narrow, and it is found only in the central and western parts of Wufeng County, Hubei Province, and the Three Gorges area, where it is endemic [17]. Moreover, due to excessive deforestation by humans and the fragility of its own genetic resources, the *M. wufengensis* population is small, its habitat is severely fragmented, and it is very scarce in the wild, with fewer than 2000 trees [18]. According to the standards of the IUCN (International Union for the Conservation of Nature), *M. wufengensis* was endangered at a level of EN A2c, which indicates that it is endemic to China [17]. Thus, it is in a critically endangered state and is in urgent need of protection. Introduction and cultivation are effective ways to protect and preserve germplasm resources of endangered plants [19]. Introduction and cultivation require an understanding of the environmental conditions that affect its growth and the selection of suitable habitat areas to be successful [20]. However, there are still some problems that need to be considered in the introduction and cultivation planning. One is the lack of detailed national growth data on *M. wufengensis*, and it is difficult to determine the range of suitable habitats across the country. Due to the huge ornamental value and economic value of *M. wufengensis*, many nurseries in China blindly follow trends and introduce species in some areas that are not suitable, resulting in poor and slow growth of *M. wufengensis*, which has caused large setbacks in the early stage [21]. The second is how to adapt to global climate change. Global climate change has become one of the main threats to rare and endangered plants in the future [22]. Future climate change will cause changes in temperature and water conditions in plant growth areas, which will significantly affect the distribution pattern of plants [23].

A viable solution to address the aforementioned problems is the utilization of species distribution models (SDMs), which involve integrating species existence data with environmental information to simulate and predict habitats suitable for species growth while also mapping the distribution of potential suitable habitats for species across space and time [24,25]. Because SDMs can reveal the relationship between the suitability of the habitat and the environment for species, they are widely used in ecological research [26], especially in predicting the distribution of species and their hotspots [27], managing invasive species [28], protecting of endangered species [11], and verifying the relationship between climate change and species distribution [29]. The commonly used SDMs mainly include the CLIMEX model, Genetic Algorithm for Ruleset Production (GARP), MaxEnt model, Bioclim model, and Domain model [30–32]. Each mode is based on different principles and algorithms, and the applicable conditions are also quite different [33,34].

While some studies have focused on the potential suitable habitat areas of *M. wufengensis* in the early stages [21,35], there was no model screening and a lack of potential habitat simulation at the Chinese scale. The objective of this study is to employ an SDM to predict the potential habitat distribution of *M. wufengensis* in China, considering the implications of future climate change. We believe that the SDMs should be screened first, and the most suitable model can be selected to effectively predict its potential suitable habitat distribution. Meanwhile, as China is the global distribution center of Magnolia plants, there are related species distributed from south to north [36]. We hypothesize that *M. wufengensis*, a newly discovered species of the genus Magnolia, can grow in most parts of China. With global warming in the future, the north-south boundary of China's climate may move northward [25]. According to the prediction results of most sympatrically distributed species [37,38], this may also cause the distribution range of *M. wufengensis* to move northward or even expand as a whole.

Therefore, this study aims to utilize four ecological niche models, Maxent, GARP, Bioclim, and Domain, all of which can be constructed solely with species occurrence point data, to predict the potential habitat of *M. wufengensis* in China. These four models have demonstrated the ability to yield reliable predictions even when distribution sample data and environmental variables are limited [33]. Specifically, we aimed to answer the following questions: (1) Which SDMs best simulate the potential habitat of *M. wufengensis* in China? (2) What are the main environmental factors affecting the habitat distribution of *M. wufengensis* in China? What is the threshold? (3) Where is the potential habitat distribution of *M. wufengensis* under future climate change? How has it changed from the current distribution? Our study will provide scientific guidance for the response to climate change, ex situ conservation, resource protection, precise introduction, cultivation, and rational site selection of *M. wufengensis* in China.

## 2. Materials and Methods

The method of this study consists of five major steps: (1) collecting and processing occurrence data and environmental factors from different sources; (2) modeling and comparing suitable habitat, the area under the receiver operating characteristic (ROC) curve (AUC) and the Kappa value with Maxent, GARP, Bioclim, and Domain; (3) simulating suitable habitat based on the best model under current and future climate change; (4) evaluating and exploring dominant environmental factors and thresholds; and (5) calculating the centroid of the suitable area under different climate scenarios and exploring the shift of the centroid. The step-by-step flow chart is shown in Figure 1.

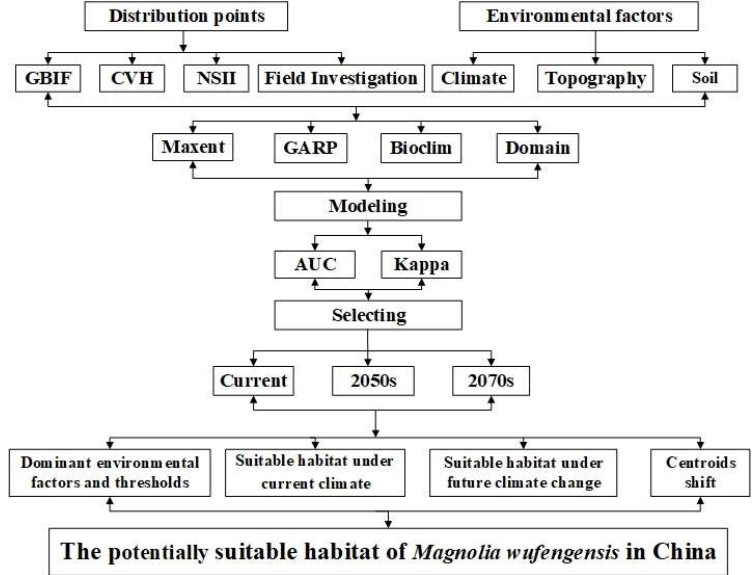

**Figure 1.** Flow chart for predicting potential suitable habitat.

## 2.1. Species Distribution Points

The distribution point data of *M. wufengensis* were obtained from data collection by the (1) Global Biodiversity Information Facility (GBIF, https://www.gbif.org/, accessed on 15 July 2019), (2) Chinese Virtual Herbarium (CVH, http://v5.cvh.org.cn/, accessed on 15 July 2019), (3) National Specimen Information Infrastructure (NSII, http://mnh.scu.edu.cn/, accessed on 15 July 2019), and (4) a field investigation (direct observation combined with GPS (Global Positioning System) technology) [21] of the main *M. wufengensis* introduction areas in China from June 2016 to October 2018 (Figure 2). After removing incorrect and duplicate distribution points and retaining distribution points spaced at least 5 km apart from adjacent points, a total of 49 *M. wufengensis* distribution record points were obtained.

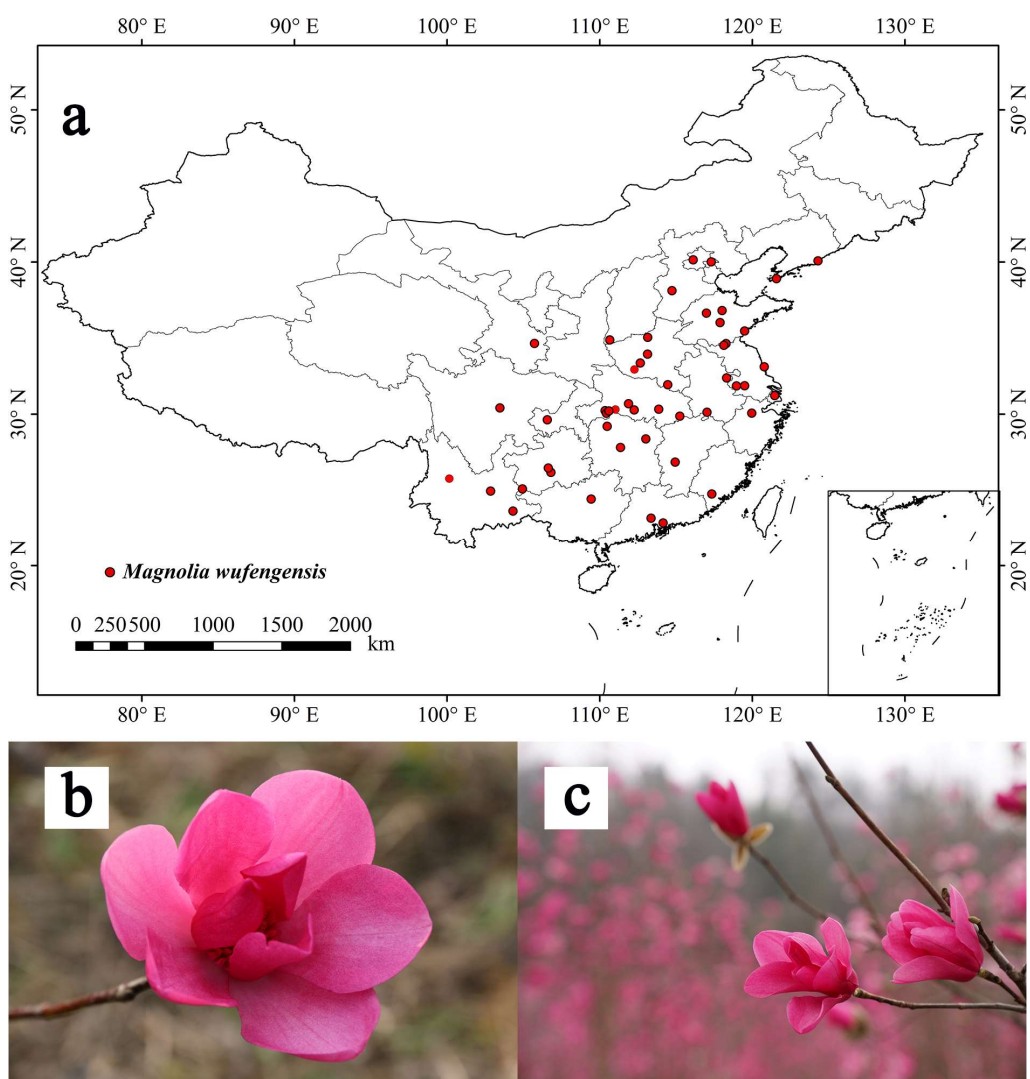

**Figure 2.** (**a**) Distribution points of *M. wufengensis* in China; (**b**,**c**) *M. wufengensis*.

## 2.2. Environmental Factors

We downloaded 19 climatic factors and 3 topographical variables (elevation, aspect, and slope) for 3 periods (a current period and the future periods of the 2050s (2040–2060) and 2070s (2060–2080)) from the World Climate Database (http://www.worldclim.org/, accessed on 15 July 2019) (Table S1). Here, we chose the CCSM4 climate system model, which has great advantages in climate simulation [39]. For the future climate data in the 2050s and 2070s, we selected four RCPs: RCP2.6, RCP4.5, RCP6.0, and RCP8.5.

We downloaded 11 soil factors from the Harmonized World Soil Database (HWSD, http://www.fao.org/soils-portal/soil-survey/soil-maps-anddatabases/harmonized-world-soil-database-v12/en/ accessed on 15 July 2019) (Table S1).

To reduce the complexity of the model and prevent overfitting of the model due to the correlation among various environmental factors, which affect the accuracy of the model [40,41], we used R software (Version 3.6.1) to perform Pearson's analysis and selected environmental factors with a correlation lower than 0.8.

Finally, 9 climatic factors, 3 topographic factors, and 4 soil factors were chosen to simulate the suitable habitat of *M. wufengensis* [21] (Table 1).

**Table 1.** Environmental factors were ultimately selected to simulate suitable habitat.

| Category | Variable | Description | Unit |
|---|---|---|---|
| Climate | bio1 | Annual mean temperature | °C |
| | bio2 | Mean diurnal range (mean of monthly (max temp–min temp)) | °C |
| | bio6 | Min temperature of the coldest month | °C |
| | bio10 | Mean temperature of the warmest quarter | °C |
| | bio11 | Mean temperature of the coldest quarter | °C |
| | bio12 | Annual precipitation | mm |
| | bio13 | Precipitation of the wettest month | mm |
| | bio14 | Precipitation of the driest month | mm |
| | bio15 | Precipitation seasonality (coefficient of variation) | / |
| Topographic | Elevation | | m |
| | Slope | | ° |
| | Aspect | | rad |
| Soil | t-bulk | Topsoil bulk density | $kg/dm^3$ |
| | t-ph | Topsoil pH ($H_2O$) | / |
| | t-clay | Topsoil clay fraction | % |
| | t-oc | Topsoil organic carbon | % |

Environmental factors from different sources were resampled in ArcGIS to ensure a consistent spatial resolution (2.5 arcminute resolution) of the data and convert the data into a format that can be recognized by all the SDMs [42]. The base map for data processing was obtained from the 1:4,000,000 China administrative division map of the standard map service of the Ministry of Natural Resources of the People's Republic of China (http://bzdt.ch.mnr.gov.cn/, accessed on 15 July 2019).

### 2.3. Model Simulation

We used MaxEnt (MaxEnt version 3.4.1) [34], GARP models (Desktop-GARP version 1.1.6) [43], Bioclim, and Domain (based on DIVA-GIS 7.5) [44,45].

Using the "Sample Points" tool in DIVA-GIS software, 75% of the known distribution points were designated as the training data, while the remaining 25% were combined with randomly selected background points, totaling ten times the size of the distribution points, to form the testing data. To compare the distinctiveness of predictions generated by four different models, ten sets of training data and corresponding testing data were randomly generated. The training datasets were utilized for model predictions, while the testing datasets were used for model validation [46].

Upon converting 16 environmental factor datasets into the required format for each model, they were imported into the respective models, and the previously generated training data were also imported, along with the configuration of relevant parameters. The Maxent model's setup followed the guidelines outlined by Phillips et al. (2006) [34], the GARP model was configured based on Anderson et al. (2003) [43], the Bioclim settings were derived from Booth et al. (2013) [44], and the Domain model's configuration referred to Carpenter et al. [45].

Maxent model procedure: First, 16 environmental data files (.asc) were loaded into the Maxent software through the "Browse" function. Second, the training data of *M. wufengensis*

were imported into the Maxent model. Finally, we selected the "Response Curves" and "Jackknife Test", which are used to analyze the environmental factors affecting the distribution of *M. wufengensis*. For the rest, we adopted the default settings.

Desktop-GARP procedure: First, 16 environmental data files (.asc) were processed by the "Dataset Manager" in Desktop-GARP, converted into a format recognized by Desktop-GARP (.raw), and loaded into the software in the form of a dataset. Second, the training data of *M. wufengensis* (Upload Data Points) were loaded into the Desktop-GARP software. The default parameter settings were selected, all 4 genetic rules were selected, the model was run 100 times, the maximum number of iterations was 1000, and the convergence limit was 0.01. Due to the instability of the model operation, according to the method of Anderson et al. (2003) [43], the "best subsets" were enabled, and the internal testing features were activated to select the 10 best models, which were added and superimposed in ArcGIS. A final grid map with a range value of 0–10 was obtained, i.e., the predicted potential distribution map of *M. wufengensis*, and the grading calculation was then performed.

Both the Bioclim and Domain models were simulated based on DIVA-GIS. First, the 16 environmental data files (.asc) were converted into a format recognized by DIVA-GIS (.raw), and the stack dataset was then generated. Second, the training data of *M. wufengensis* were inserted into "Data". Finally, the environmental dataset was added in stack format to the Modeling-Bioclim/Domain module, and predictions of the Bioclim and Domain models were then generated.

### 2.4. Model Evaluation

In this study, the accuracy of the model was evaluated using the area under the curve (AUC) and Cohen's kappa. The AUC value is widely employed due to its threshold-independent nature, making it a robust measure for model performance assessment [47,48]. The prediction results of each model were converted into "grd" format in DIVA-GIS. Then, the testing datasets were imported, the evaluation file was created, and the AUC and Kappa values were output [49].

The range of the AUC is between 0 and 1. An AUC < 0.7 suggests that the prediction performance is extremely poor, values between 0.7 and 0.8 indicate moderate performance, values between 0.8 and 0.9 suggest good performance, and values between 0.9 and 1.0 indicate excellent performance [50]. In other words, the closer the value is to 1, the better the model fit. The evaluation criteria for Kappa were excellent, 1.0–0.81; very good, 0.80–0.61; good, 0.60–0.41; fair, 0.40–0.21; and fail, <0.20 [51,52].

### 2.5. Suitable Habitat Partitions under Current and Future Conditions

According to this study, which was repeated 10 times, each model produced 10 sets of prediction results. The prediction result with the largest AUC for each model was selected as the base map, and ArcGIS was used for raster format conversion and reclassification. The species existence probabilities derived from each model were used to classify the predicted potential suitable area for *M. wufengensis* of the different models, as shown in Table 2, and the suitable habitat distribution maps of the different models were then obtained [43,52].

**Table 2.** Classification standards of suitable habitats for different SDMs.

| Suitability | GARP | Maxent | Bioclim | Domain |
|---|---|---|---|---|
| Unsuitable | [0, 2] | [0, 0.2] | 0 | [0, 90] |
| Lowly suitable | (2, 4] | (0.2, 0.4] | (0, 5%] | [91, 93] |
| Moderately suitable | (4, 6] | (0.4, 0.6] | (5%, 10%] | [94, 96] |
| Highly suitable | (6, 10] | (0.6, 1.0] | (10%, 27%] | [97, 100] |

## 3. Results

### 3.1. Prediction Results of Four Models

In this study, four SDMs were utilized to predict the distribution map of the current potential suitable habitat for *M. wufengensis* in China (Figure 3). The representation of

highly suitable, moderately suitable, lowly suitable, and unsuitable habitats is denoted by the red, yellow, green, and white areas, respectively (the same applies below).

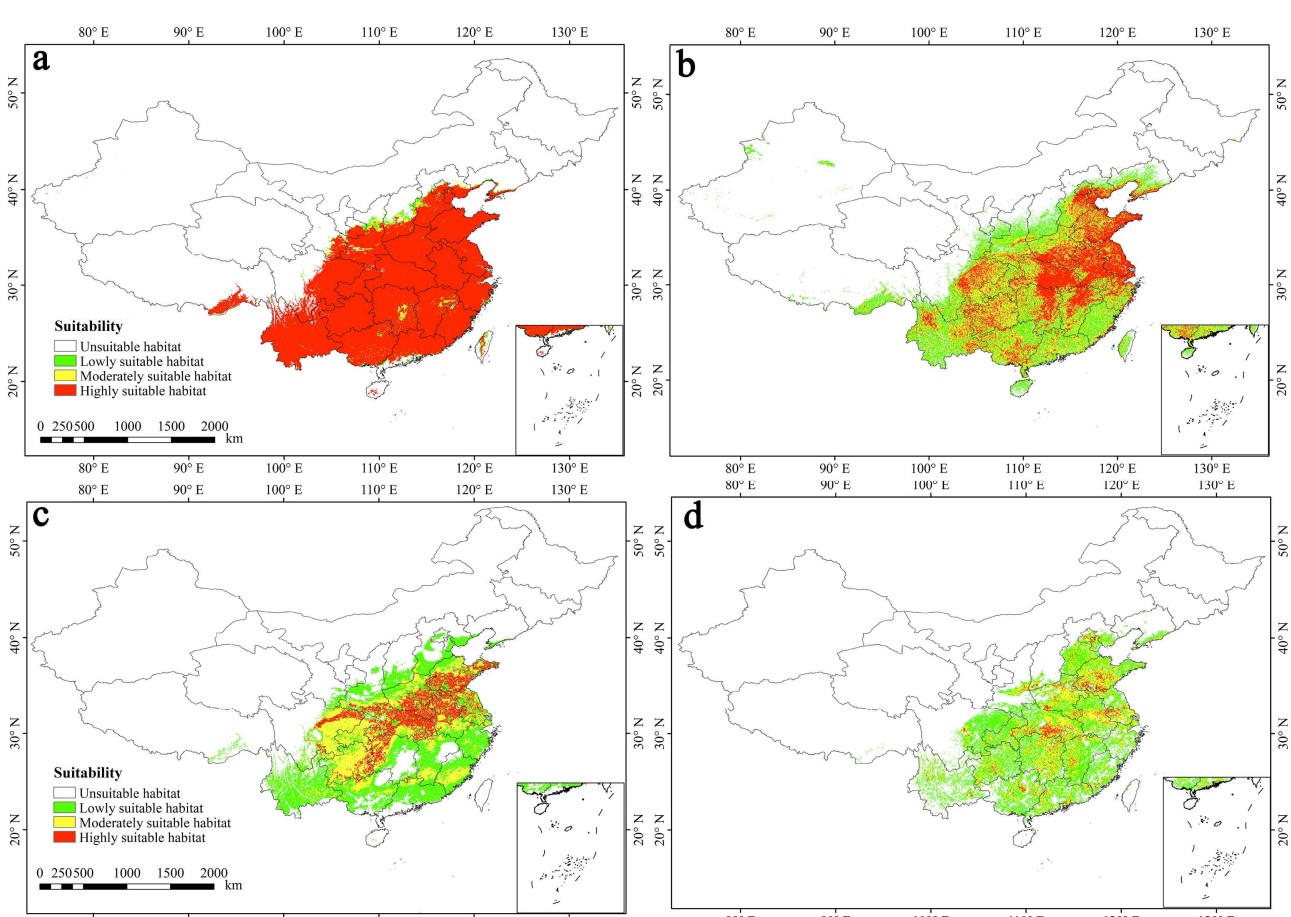

**Figure 3.** Distribution map of suitable habitat for *M. wufengensis* in China simulated by four SDMs ((**a**) GARP; (**b**) Maxent; (**c**) Bioclim; and (**d**) Domain).

The suitable habitats of *M. wufengensis* predicted by the four SDMs were all distributed south of 40° N and east of 90° E in China. In terms of climate zone, the potential suitable habitats were mainly in the subtropical temperate and humid monsoon climate zones, and some areas were in the temperate monsoon climate zone, mainly including Hebei, Shandong, Henan, Anhui, Jiangsu, Hubei, Hunan, Zhejiang, Jiangxi, Fujian, Guangdong, Guangxi, Yunnan, Chongqing, Guizhou, and parts of eastern Sichuan, Shaanxi, Shanxi, Gansu, and Liaoning.

Due to the different principles and algorithms of the different models, there were some differences in their results. The distribution map of the suitable habitats for *M. wufengensis* obtained by the GARP simulation (Figure 3a) showed that most of the suitable habitat was highly suitable, while there were very limited moderately and lowly suitable habitats. The range of suitable habitats of different grades for *M. wufengensis* simulated by the Maxent model was relatively distinct (Figure 3b) and was similar to Bioclim's prediction (Figure 3c). The range of highly suitable habitats for *M. wufengensis* simulated by the Domain model was very small and scattered (Figure 3d), and most of the suitable habitats were lowly suitable habitats.

The areas of different grades of suitable habitats for *M. wufengensis* simulated by the four models in Figure 4 show that the GARP model simulated the largest area of highly suitable habitat at $273.05 \times 10^4$ km$^2$, which was 2.86 times that simulated by the Maxent model ($95.26 \times 10^4$ km$^2$), 6.90 times that simulated by the Bioclim model ($39.53 \times 10^4$ km$^2$), and 31.00 times that simulated by the Domain model ($8.81 \times 10^4$ km$^2$). The GARP model

similarly simulated the moderate and low suitability habitats for *M. wufengensis* as the smallest areas, at $5.96 \times 10^4$ km$^2$ and $4.95 \times 10^4$ km$^2$, respectively, which was 5.63% and 5.58% of the moderately and lowly suitable habitats simulated by the Maxent model, 9.00% and 4.45% of those simulated by the Bioclim model, and 9.18% and 4.72% of those simulated by the Domain model, respectively. In general, the GARP model predicted the largest range of suitable habitats for *M. wufengensis*. The suitable habitat of *M. wufengensis* obtained by the Domain simulation was the smallest and most scattered. The prediction range of the Maxent model was closer to that of the Bioclim model, the suitable habitats of different grades were also clearer, and the distribution was more reasonable.

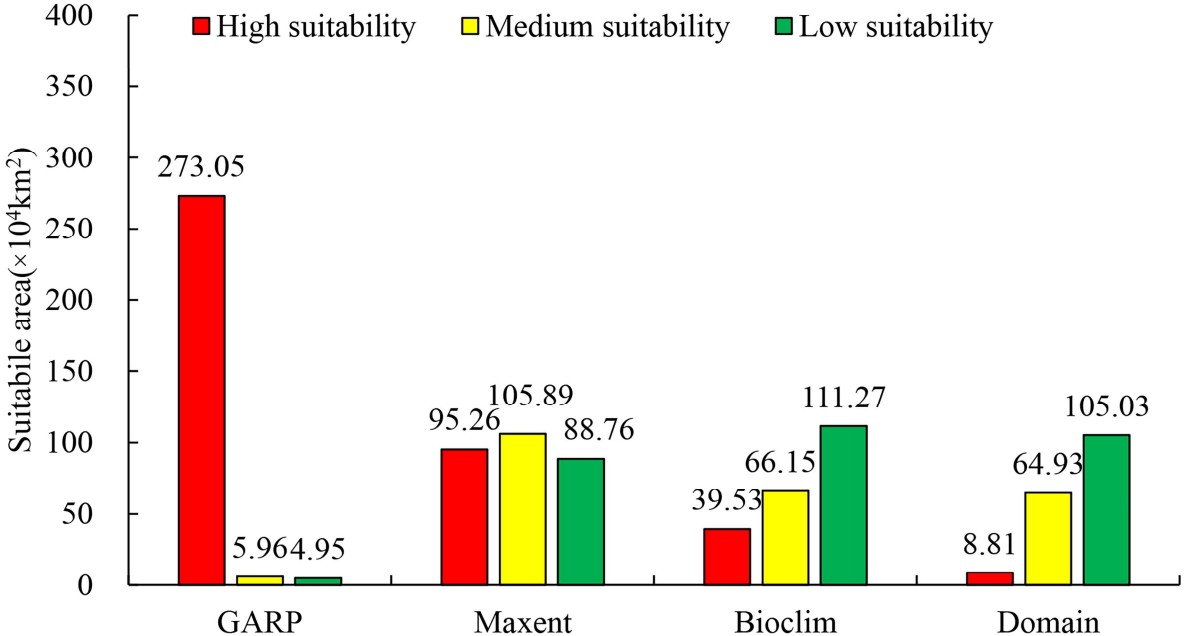

**Figure 4.** The suitable habitats for *M. wufengensis* of different grades in China simulated by four schemes.

### 3.2. Model Accuracy Evaluation

The four SDMs simulated the mean values of the AUC and Kappa for the current potential suitable habitat for *M. wufengensis* in China, as shown in Figure 5. Figure 5 shows that the average AUC of the prediction results of the four models is above 0.85, which far exceeds that of a random model (AUC = 0.5), indicating that the four models have a relatively good predictive effect for the suitable habitat of *M. wufengensis*. The standard deviation of the AUC values of the four models was ordered as follows: Domain > Bioclim > Maxent > GARP. The results of this study showed that for the suitable habitat of *M. wufengensis*, the Maxent model ($0.9479 \pm 0.0080$) had the best predictive results, followed by the Domain ($0.9367 \pm 0.0287$) and GARP ($0.8719 \pm 0.0039$) models, and the Bioclim ($0.8513 \pm 0.0177$) model had the lowest level of prediction. Figure 5 shows that the average Kappa values of the four SDMs are above 0.7, and the consistency of the models is significant. The four SDMs predicted the potential suitable habitats for *M. wufengensis* with high accuracy, and they can be used to predict the distribution of potential suitable habitats. The Kappa values were in the following order: Maxent ($0.8113 \pm 0.0228$) > Domain ($0.7629 \pm 0.0531$) > Bioclim ($0.7166 \pm 0.0372$) > GARP ($0.6969 \pm 0.0200$). In summary, combined with the AUC and Kappa values of the model, the Maxent model best predicted the potential suitable habitat for *M. wufengensis*, with the most significant consistency Table 3.

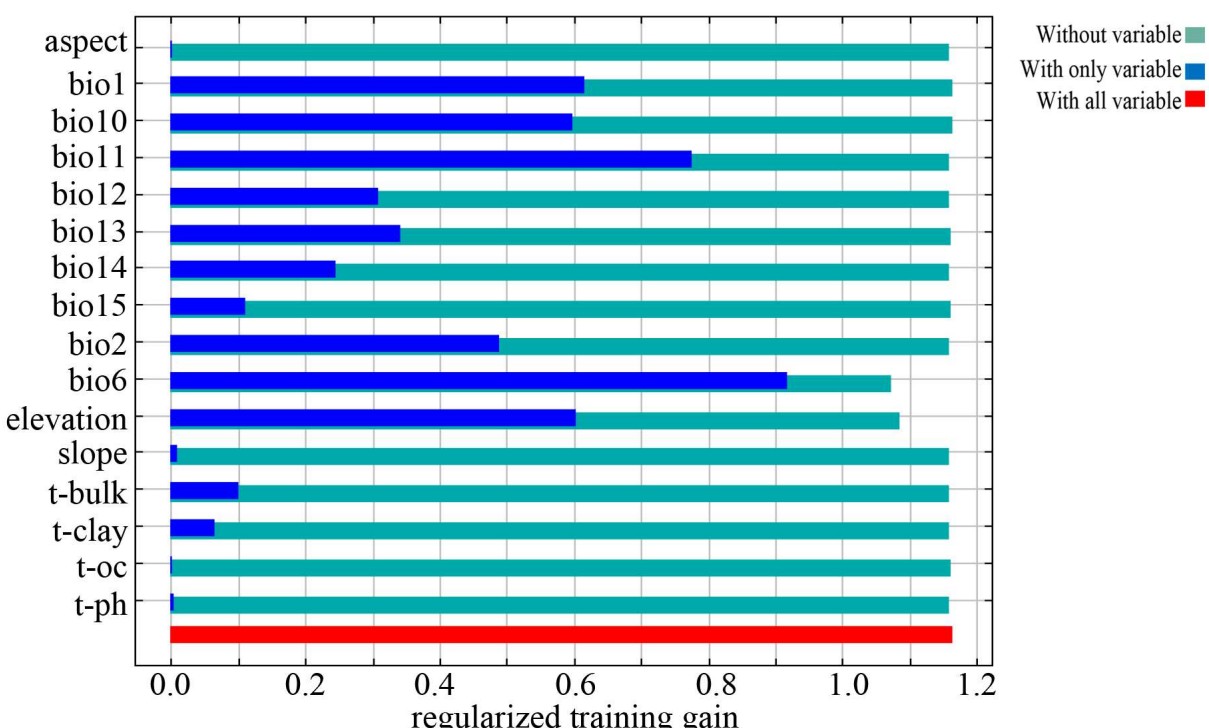

**Figure 5.** Jackknife test for the importance of the variables.

**Table 3.** Comparison of AUC and Kappa values of the results of the four SDMs. Different small letters indicate significant differences among treatments as assessed by Duncan's test ($p < 0.05$).

| Model | Maxent | GARP | Bioclim | Domain |
|---|---|---|---|---|
| AUC | $0.9479 \pm 0.0080$ a | $0.8719 \pm 0.0039$ b | $0.8512 \pm 0.0177$ c | $0.9367 \pm 0.0287$ a |
| Kappa | $0.8113 \pm 0.0228$ a | $0.6969 \pm 0.0200$ c | $0.7166 \pm 0.0372$ c | $0.7629 \pm 0.0531$ b |

### 3.3. Evaluation of Environmental Factors

Through the above analysis, it can be concluded that the Maxent model is the best model for predicting the potential suitable habitats of *M. wufengensis*. When simulating the potential suitable habitat for a species, adopting default parameters is customary, but it will cause overfitting, reduce the accuracy of the research results, and directly affect the transferability of the model [53–55]. Therefore, in this study, we used the Maxent model with parameters optimized by the ENMeval package in R software to simulate the potential suitable habitats of *M. wufengensis* in China. The relevant parameters were the regularization multiplier value (RM) of 3.5, and the feature combination (FC) was LQ (Table S2). At this time, the model fits the species distribution points well and has significantly decreased the complexity and reduced the degree of overfitting.

After predicting the potential suitable habitat of *M. wufengensis* through the Maxent model with optimized parameters, the percent contributions and the permutation importance of each environmental factor were determined (Table S3). Among the 16 environmental factors, the minimum temperature of the coldest month (bio6), elevation (28.73%), and mean temperature of the coldest quarter (bio11) ranked in the top 3, with cumulative values as high as 96.07% and 92.32%, respectively (Table S3).

According to the results of the jackknife test (Figure 5), using factors other than the ones examined, the regularized training gain increased the most in bio6, followed by bio11 and bio1.

The dominant environmental factors for the potential suitable habitats of *M. wufengensis* in this study were bio6, bio11, bio1, and elevation.

The response curve (Figure 6) shows that when the distribution probability of suitable habitats for *M. wufengensis* was greater than the threshold (0.2), the intervals of the dominant environmental factors restricting the distribution of *M. wufengensis* were bio6 (−13.36–9.84 °C), bio11 (−6.06~−12.66 °C), bio1 (≥4.49 °C), and elevation (0–2803.93 m).

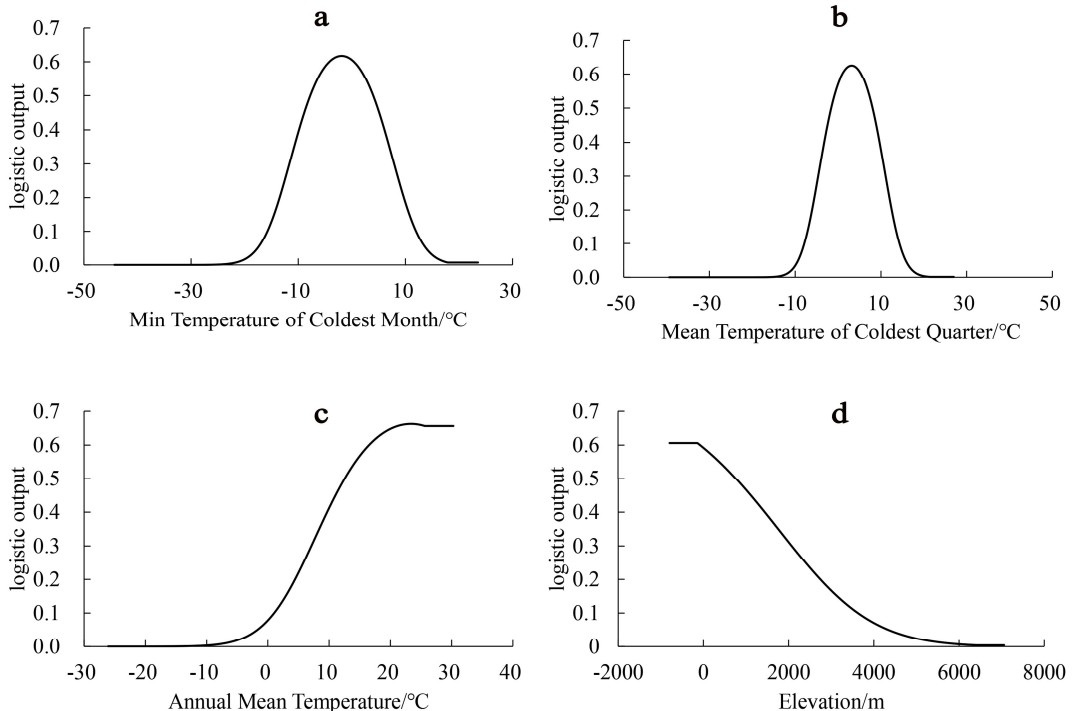

**Figure 6.** Responses of *M. wufengensis* to the four main environmental factors.

### 3.4. Current Potential Suitable Habitats in China

The results for *M. wufengensis* in the current potentially suitable habitats in China under the optimized parameters are shown in Figure 7.

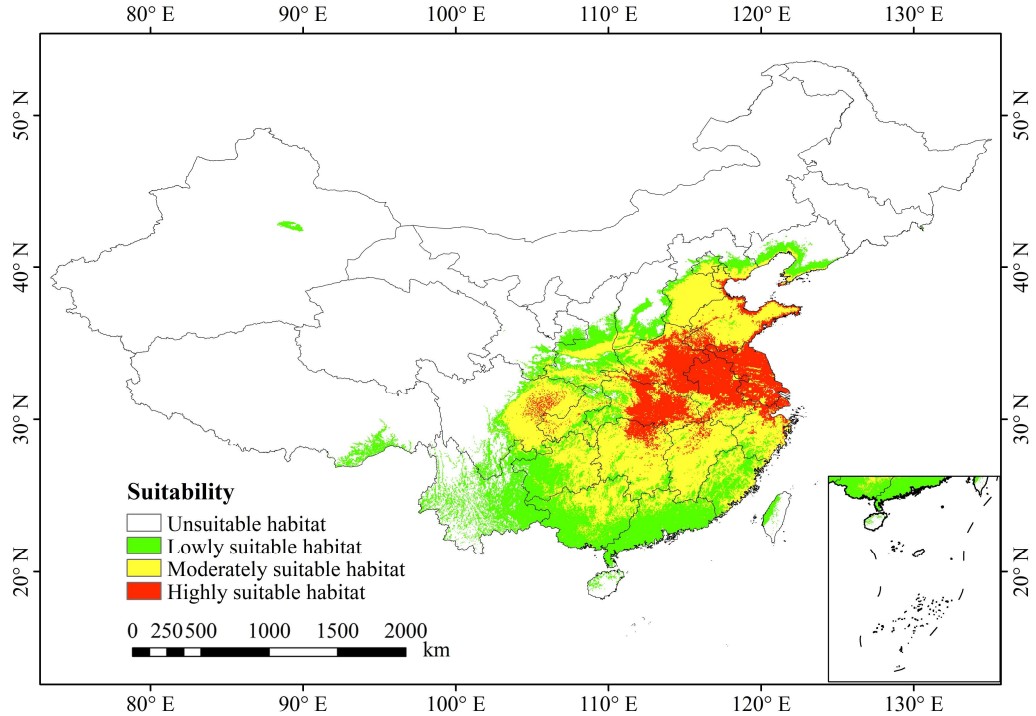

**Figure 7.** Suitable habitats for *M. wufengensis* in China under the current period.

The highly suitable habitats for *M. wufengensis* were highly concentrated and were mainly distributed in East China and Central China, covering Hubei, Henan, Anhui, Jiangsu, Shanghai, the northern part of Hunan, Jiangxi, Zhejiang, southern Shandong, and eastern Sichuan. The moderately suitable habitats mainly surrounded the highly suitable habitats, covering eastern Sichuan, Hunan, Jiangxi, Zhejiang, Fujian, Shandong, southeastern Hebei, southern Shanxi, eastern Guizhou, the northern parts of Guangxi and Guangdong, and Chongqing, Beijing, and Tianjin. The lowly suitable habitats were mainly in Yunnan, southwestern Guizhou, the southern parts of Guangxi and Guangdong, southern Fujian, central Sichuan, southeastern Gansu, the southern parts of Shaanxi and Shanxi, and small areas of Hebei and Liaoning. Currently, although Xinjiang and Tibet do not have *M. wufengensis* populations, according to the MaxEnt prediction, Turpan in Xinjiang and Linzhi in Tibet nevertheless have small areas of lowly suitable habitats, indicating that the model has high transferability. Under the current climate scenario, the areas of highly, moderately, and lowly suitable habitat for *M. wufengensis* were $46.60 \times 10^4$ km$^2$, $122.82 \times 10^4$ km$^2$, and $96.36 \times 10^4$ km$^2$, accounting for 4.85%, 12.79%, and 10.03% of China's total land area, respectively.

### 3.5. Potential Suitable Habitat and Dynamic Changes in the Future

Four climate change scenarios (RCP2.6, RCP4.5, RCP6.0, and RCP8.5) in the 2050s and 2070s were selected, and the optimized MaxEnt model was used to simulate the distribution of potential suitable habitat for *M. wufengensis* (Figure 8), the dynamic changes relative to the current suitable habitat (Figure 9), and the changes in area (Table 4) under future climate change conditions.

**Table 4.** Suitable habitat area change for *M. wufengensis* under different climate change scenarios.

| Climate Scenario | High Suitability | | Medium Suitability | | Low Suitability | | No Suitability | |
|---|---|---|---|---|---|---|---|---|
| | Area (10$^4$ km$^2$) | Change Rate (%) | Area (10$^4$ km$^2$) | Change Rate (%) | Area (10$^4$ km$^2$) | Change Rate (%) | Area (10$^4$ km$^2$) | Change Rate (%) |
| Current | 46.59 | / | 122.82 | / | 96.36 | / | 694.83 | / |
| RCP2.6 (2050) | 32.63 | −29.98 | 130.74 | 6.45 | 117.43 | 21.86 | 679.81 | −2.16 |
| RCP4.5 (2050) | 33.48 | −28.15 | 154.86 | 26.09 | 97.10 | 0.77 | 675.16 | −2.83 |
| RCP6.0 (2050) | 36.34 | −22.00 | 156.09 | 27.09 | 92.26 | −4.25 | 675.91 | −2.72 |
| RCP8.5 (2050) | 37.85 | −18.77 | 144.75 | 17.86 | 101.48 | 5.31 | 676.53 | −2.63 |
| RCP2.6 (2070) | 33.58 | −27.94 | 147.68 | 20.24 | 103.54 | 7.45 | 675.81 | −2.74 |
| RCP4.5 (2070) | 34.01 | −27.02 | 149.15 | 21.44 | 102.21 | 6.07 | 675.23 | −2.82 |
| RCP6.0 (2070) | 36.65 | −21.33 | 143.50 | 16.84 | 100.99 | 4.80 | 679.46 | −2.21 |
| RCP8.5 (2070) | 44.48 | −4.54 | 144.27 | 17.46 | 99.31 | 3.06 | 672.55 | −3.21 |

The results showed that the potential suitable habitat of *M. wufengensis* will undergo various changes under future climate change scenarios. The highly suitable habitats for *M. wufengensis* first decreased and then increased with increasing RCP value and time, but the area was smaller than the current area and exhibited serious fragmentation in the 2050s and 2070s (Figures 8 and 9). Among the various predictions, the highly suitable habitats were the smallest at only $32.62 \times 10^4$ km$^2$ under RCP2.6 in the 2050s, which was 29.98% lower than the current area (Table 4), and the areas that exhibited the main reductions were in southwestern Shandong, central Henan, the northern part of Anhui and Jiangsu, which are characterized by plains and would become moderately suitable habitats (Figure 9a). Under the RCP8.5 scenario in the 2070s, the highly suitable habitats were the closest to those of the current period (Figures 8h and 9h; Table 4). However, under future climate change, the highly suitable habitat for *M. wufengensis* increased significantly and was concentrated in the Sichuan Basin (Figure 8).

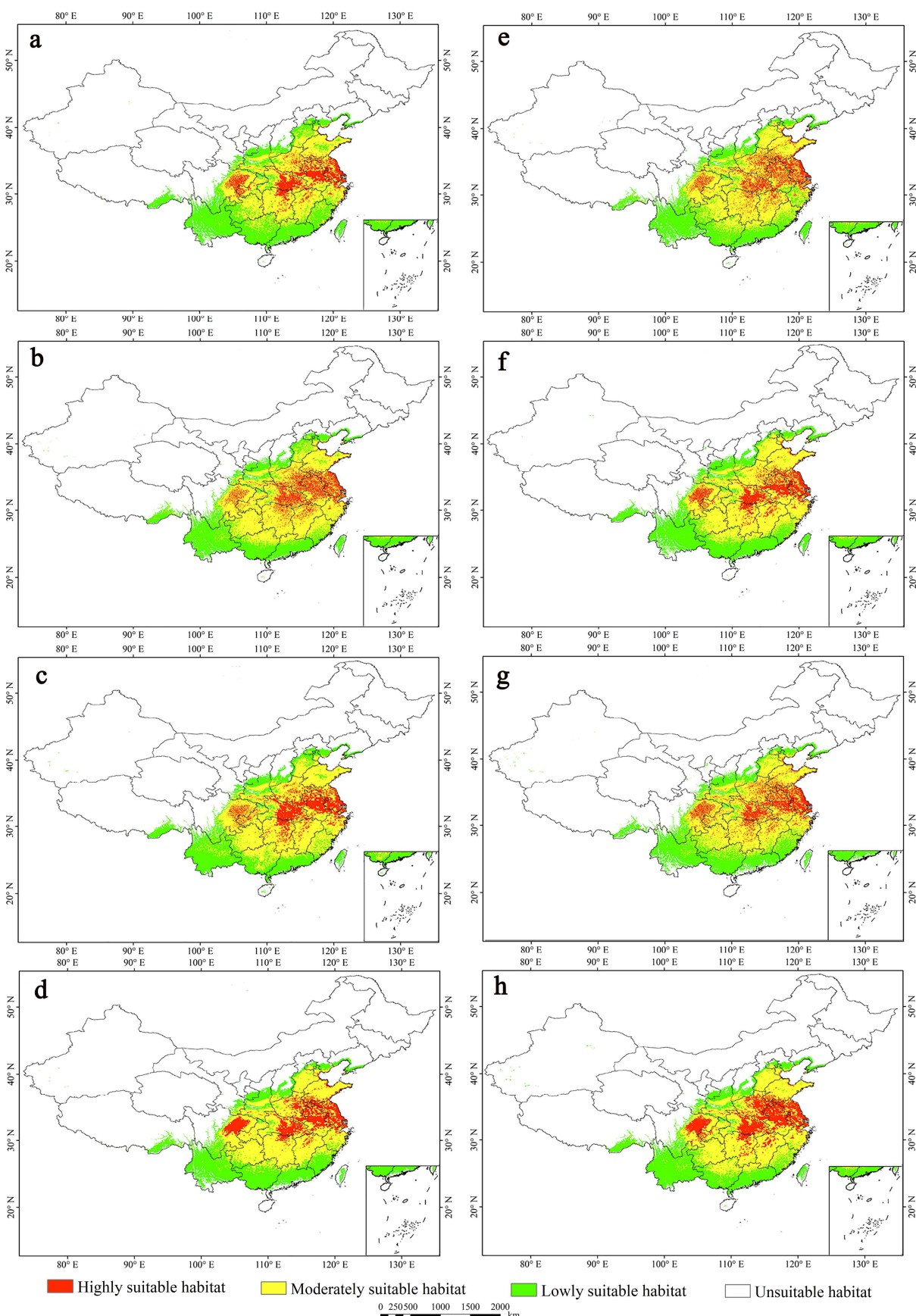

**Figure 8.** Distribution of suitable habitats for *M. wufengensis* under future climate change scenarios ((**a**–**d**) 2050s: RCP2.6, RCP4.5, RCP6.0, and RCP8.5; (**e**–**h**) 2070s: RCP2.6, RCP4.5, RCP6.0, and RCP8.5).

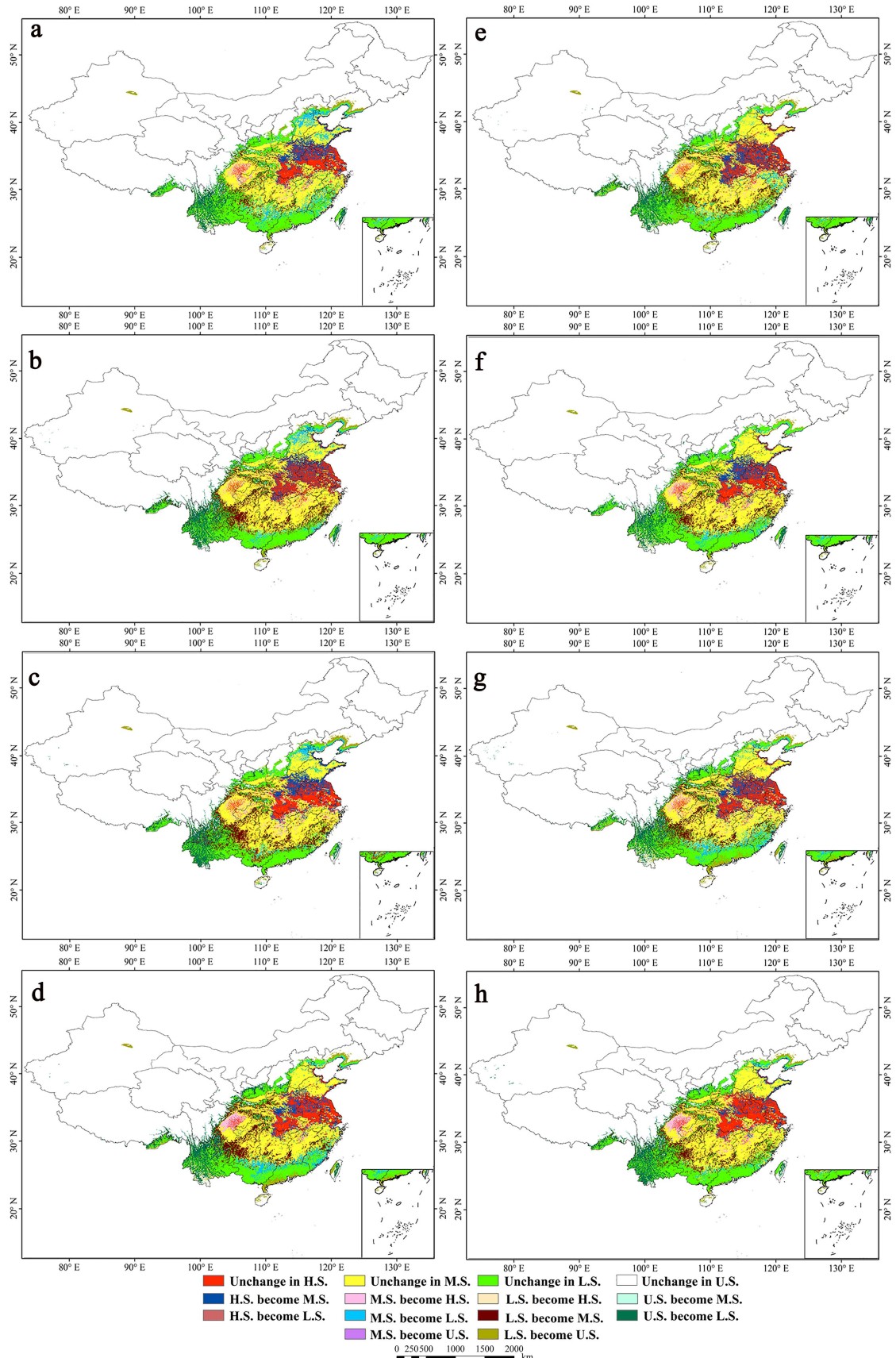

**Figure 9.** Dynamic changes in potential suitable habitats for *M. wufengensis* under future climate change scenarios ((**a**–**d**) 2050s: RCP2.6, RCP4.5, RCP6.0, and RCP8.5; (**e**–**h**) 2070s: RCP2.6, RCP4.5, RCP6.0, and RCP8.5).

Moderately suitable habitats for *M. wufengensis* under future climate change will increase compared to those of the current period (Table 4). Under the RCP2.6 scenario in the 2050s, the increased area was the smallest at only 6.45% (Table 4). The area of moderately suitable habitats reached a maximum of 156.09 × 10$^4$ km$^2$ under RCP6.0 in the 2050s, which was 27.09% larger than the current area, and the areas of increase were mainly in Shandong, southern Henan, Jiangsu, northern Anhui, the southwestern part of Guizhou, and the central part of Guangxi. There was a trend of expansion toward highly and lowly suitable habitats (Figures 8c and 9c).

The lowly suitable habitat for *M. wufengensis* showed a trend of first increasing and then decreasing under the RCP2.6 and RCP8.5 scenarios in both the 2050s and 2070s and gradually increasing under the RCP4.5 scenario, while the suitable habitats first decreased and then increased under the RCP6.0 scenario (Table 4). Under the RCP2.6 scenario in the 2050s, the lowly suitable habitat reached a maximum area (117.43 × 10$^4$ km$^2$), which was 21.86% higher than the current area (Table 4). The areas of increase were mainly in southeastern Tibet and most of Yunnan and Taiwan Island, which tended to extend toward moderately suitable and unsuitable areas (Figures 8a and 9a), while under the RCP6.0 scenario in the 2050s, the area of lowly suitable habitat was 4.25% less than the current area, with little change (Figure 9c). In the future, under different climate scenarios, the increased areas will mainly be high-elevation mountain areas. For example, the area of the lowly suitable habitat for *M. wufengensis* on Taiwan Island will increase and move to the central high-elevation area, and on Hainan Island, this area will gradually decrease to only central high-elevation areas (Figures 8 and 9).

Overall, the area of suitable habitats for *M. wufengensis* will decrease under future climate change scenarios, with an overall decrease of 4.53%–29.98%; the area of moderately suitable habitats will continue to increase in the future, with an increase of 6.45%–27.09%; the area of lowly suitable habitats will increase by 0.77%–21.86%; and the area of unsuitable habitats will gradually decrease by 2.16%–3.21% in the future.

### 3.6. Centroid Shifts in Different Suitable Habitats

Figure 10 indicates that the centroids of the highly suitable habitats (Figure 10a) and the moderately suitable habitats for *M. wufengensis* (Figure 10b) migrate mainly to the northeast under future climate change (Figure 10(a1,b1)). The centroids of the lowly suitable habitats (Figure 10c) migrate mainly northward (Figure 10(c1)). In general, the centroid of the suitable habitats for *M. wufengensis* moves more to the northeast.

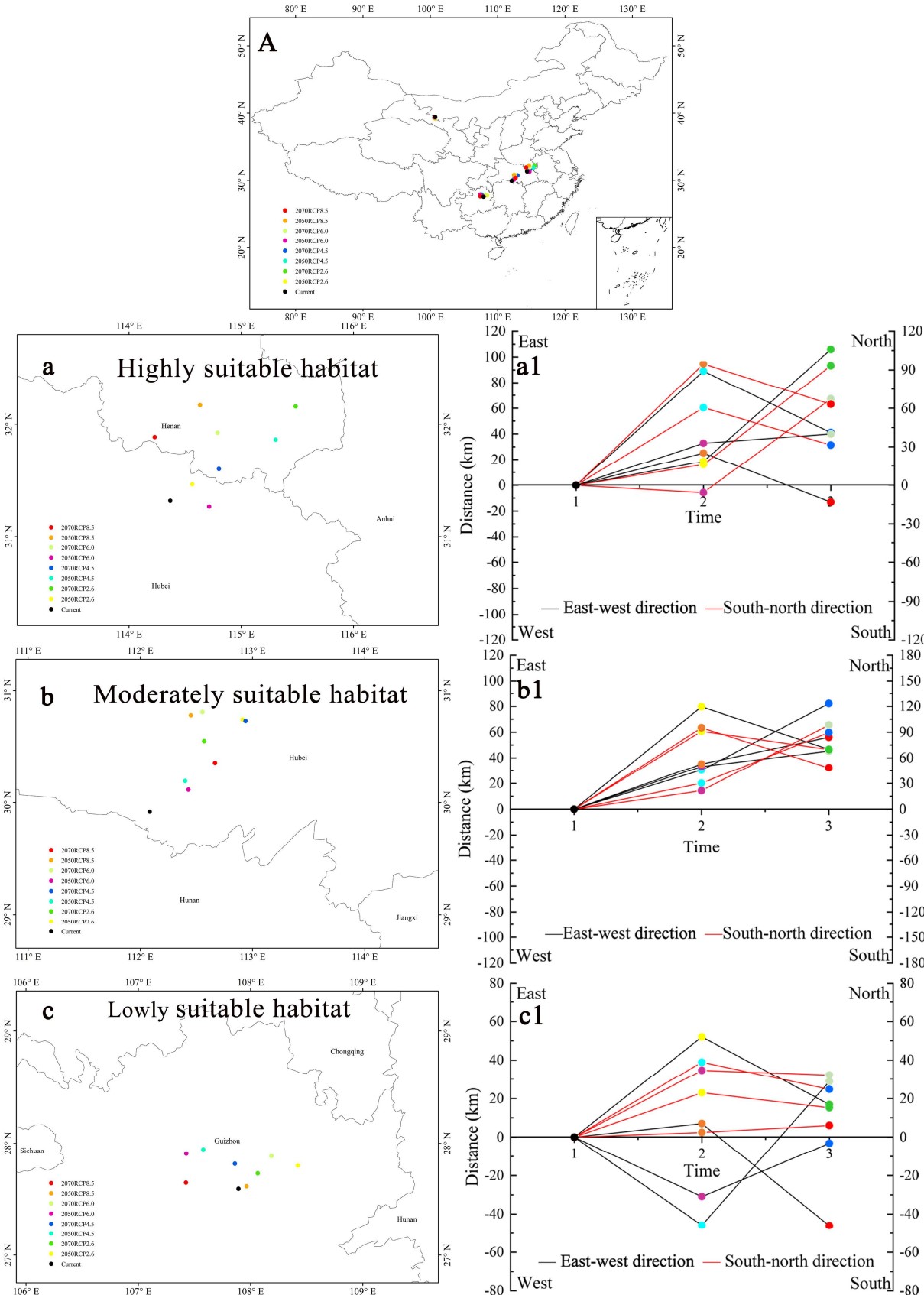

**Figure 10.** Centroid migration in suitable habitats for *M. wufengensis* under future climate scenarios in China. (**a1–c1**) indicates the distances that centroids of different grades of suitable habitat migrate in two directions (north–south and east–west) under future climate change.

## 4. Discussion

### 4.1. Model Performance

*M. wufengensis* is an endangered and rare tree species in China. To provide better protection for it, we used four SDMs to study its suitable habitat area. While the four SDMs employed in this study demonstrated promising outcomes in simulating the distribution of potential suitable habitats for *M. wufengensis*, some variations were observed in the predicted suitable habitats among the different models. This is mainly because the different SDMs are based on related mathematical algorithms and fit data in different ways [56]. In this study, the GARP model exhibited the broadest range and the largest area of suitable habitats predicted for *M. wufengensis*. This can be attributed to the GARP model's foundation on the principle of the genetic rule algorithm [57], which undergoes continuous iteration to screen and evaluate the dataset. By simulating the ecological requirements of the species, the model effectively determines the potential suitable areas within the study region. Related studies have further confirmed that the GARP model commonly faces the issue of overprediction, displaying a tendency to predict species fitness beyond their known niches [58,59], presumably due to GARP's failure to model lesser relationships in the data [60]. In addition, the GARP model's rulemaking process did not include factors that caused species to spread to these incorrect areas [61]. Compared to the GARP model, the Bioclim model exhibits a smaller overall suitable habitat area, yet with distinct levels of specificity. This distinction primarily arises from the environmental envelope principle that underpins the Bioclim model [44]. This model assumes that the environmental climate within the envelope is similar to the actual distribution area's climate, allowing a certain species to grow and reproduce normally within this environment [52]. As a result, the predictions are relatively conservative. In contrast to the GARP model, the Domain model predicts the majority of the suitable habitat as low suitability areas. This discrepancy arises from the fact that the Domain model classifies based on a similarity matrix between points [45]. This is largely influenced by sample points, leading to the appearance of certain suitable habitat areas in nearly all locations where distribution points exist [46]. With fewer sample points, the level of suitability naturally tends to be lower. In comparison, while the potential suitable habitats of *M. wufengensis* predicted by the Maxent model closely resembled those of the GARP model, the distinctions between the different habitat grades were more pronounced, resulting in clearer and more defined predicted outcomes. This is because the Maxent model does not make any assumptions about that which is unknown under the constraints of the maximum entropy theory algorithm and runs completely according to the maximum entropy of that which is closest to their actual states, so its predictions of suitable habitats for species are more accurate. Zhang et al. (2016) [62] also found that the area of suitable habitat predicted by the Maxent model was smaller than that of the GARP model for the distribution of *Arceuthobium sichuanense* in China; however, the level of local detail in their study was more distinct. Moreover, the Maxent model predicted that the suitable habitats of *M. wufengensis* can reach Yining, Bole, and Turpan in Xinjiang, which is an area that was not predicted by the other three models, reflecting the good transferability of the Maxent model.

The most important assumption of the Bioclim model is that species can grow, develop, and reproduce in places where the climatic conditions are similar or comparable to those in their current living environment [45], so the results of its predictions of species distribution are relatively conservative. The Domain model is greatly affected by the sampling of the original distribution point of the species when predicting their distribution area and will spread around according to the sampling distribution point, resulting in the expansion or contraction of the ecological niche of the species distribution area [52,60]. Thus, the results of this study about these four models are consistent with previous studies. Wang et al. (2020) [49] compared and analyzed the predictive effects of these four models when researching and predicting the potential suitable areas of *Pseudolarix amabilis* and found that the prediction range of GARP was wider, the prediction range of Bioclim was smaller, and the Maxent model demonstrated higher accuracy in its prediction results compared to the

other models. Duan et al. (2020) [63] studied the potential distribution of *Ammopiptanthus* species in China and found that the Maxent model yielded the highest AUC value and exhibited the best overall performance in predicting the species' distribution. Elith et al. (2006) [60] demonstrated that the Maxent model outperformed the Bioclim, Domain, and GARP models when utilizing 16 different SDMs to predict the distribution of potential suitable areas for 226 species.

*4.2. Dominant Environmental Factors*

The dominant environmental factors limiting the distribution of *M. wufengensis* in China were the minimum temperature of the coldest month (bio6), mean temperature of the coldest quarter (bio11), annual mean temperature (bio1), and elevation. Zhu (2012) found that extremely low temperatures and their duration were important limiting factors during the overwintering of *M. wufengensis*. Yang (2015) [64] found that during an actual introduction process in North China, *M. wufengensis* under 3–4 years old could not naturally overwinter, and they recommended that appropriate cold protection measures be taken to protect *M. wufengensis* from low-temperature freezing damage. Currently, the most advanced landscaping methods are to build windproof barriers and use insulation cotton to wrap and cover trees [65]. This observation aligns with the finding that the Northeast China region is unsuitable for the survival of *M. wufengensis*, while the North China region comprises habitats with moderate and low suitability for the species. Liang (2010) [16] reported that the semilethal temperature of *M. wufengensis* is −15 °C, which closely aligns with the lower limit of −13.36 °C derived from the response curve of bio6 in this study. Numerous researchers studying other plants with sympatric distributions have also reached similar conclusions. For instance, Yan et al. (2019) [66] found that bio11 significantly influenced the distribution of *Pinus massoniana*. Zhang et al. (2018) [38] observed that temperature exerted a substantial impact on the potential geographical distribution of *Sorbus amabilis*. Similarly, Li et al. (2016) [67] analyzed the importance of eight climate variables for *Quercus chenii* using the MaxEnt model, and their findings highlighted that the caloric index played a pivotal role in limiting the species' geographical distribution, followed by the water index.

Elevation affects the growth of *M. wufengensis* by affecting soil physicochemical properties, soil enzyme activity, light conditions, and air temperature, thereby limiting the vertical geographic distribution of the species. The upper limit of the distribution of *M. wufengensis* under natural conditions is 1400–2000 m [14,15], which is within the upper limit of 2803.93 m obtained from the elevation response curve in this study.

The results of this study showed that precipitation had little effect on the distribution of *M. wufengensis* in China. However, precipitation plays a vital role in plant growth. *M. wufengensis* is a succulent root that is sensitive to water, and it easily dies if it receives too much water [68]. The current highly suitable habitats for *M. wufengensis* are mainly in East China. These areas are also traditional subtropical monsoon climate areas in China. One of the most notable features of these areas is that rain and heat occur in the same period, and it is easy to form a high-temperature and high-humidity environment. In the actual investigation, pests and diseases were detected. The main diseases of *M. wufengensis* were sooty blotch on the surface of leaves and root rot, which causes the plants to rot from the root stem, and the main pests were sucking pests such as *Tetranychus cinnabarinus*, *Liriomyza*, and *Pseudococcus* [21]. Therefore, suitable site conditions should be considered when introducing *M. wufengensis*. For example, in low-lying areas, high beds should be made, and in the summer, cleaning and drainage should be performed as appropriate. If sooty blotch occurs, it is necessary to spray the plant with a 0.3%–0.5% mass fraction of Bordeaux mixture for prevention and control, and root rot should be controlled by continuously spraying 2–3 times with 500 times carbendazim solution, which can mostly control the development of the disease. In addition, sucking pests should be sprayed 500–1000 times with omethoate for prevention [21].

### 4.3. Current Suitable Habitats

Our study revealed that the potential suitable habitats for *M. wufengensis* are currently widespread across southeast China, with areas of high suitability observed in Hubei, Henan, Anhui, Sichuan, Zhejiang, Jiangsu, and Hunan. These results are consistent with a previous study [21], which used the fuzzy similarity priority ratio method based on the principle of climatic similarity and revealed that the suitable habitats of *M. wufengensis* were mainly distributed in most areas from the eastern Huanghuai Plain to the northern Guangdong and Guangxi Hills. According to other studies [69,70], there are 17 species of *Magnolia* in East China, 20 species in Central China, and 29 species in South China, demonstrating a high richness of *Magnolia* species in this study area and relatively reliable research results, which can provide a scientific basis for the effective protection and precise introduction and cultivation of *M. wufengensis*. Interestingly, unlike previous studies, our study showed that Turpan in central Xinjiang and Linzhi in southern Tibet are also suitable for *M. wufengensis*, providing evidence that the Maxent model has good transferability.

### 4.4. Changes to the Suitable Habitats in the Future

In the context of future global climate change, rare and endangered plants will face higher extinction risks than other plants due to their narrow habitats, sparse populations, and weak natural regeneration [71]. Understanding the potential contraction or expansion of their habitats is of great significance for the conservation of these rare and endangered species. According to our research results, although the highly suitable habitats of *M. wufengensis* show a decreasing trend under different climate change models in the future, the moderately and lowly suitable habitats show an increasing trend. This indicates that although some areas may not be suitable for the growth of *M. wufengensis*, there will be some new areas suitable for its growth, which is very beneficial for its conservation and introduction. Gao et al. (2022) [72] found that the highly suitable habitat of the rare and endangered plant *Firmiana kwangsiensis* in Guangxi is expected to decrease over time, and it will become extinct in some areas, but it can also adapt to some new areas. Li et al. (2019) [73] showed that in the case of future global warming, the potential suitable habitats of *Osmanthus yunnanensis* expand to the east and north, and the areas of suitable distribution increase; the potential suitable habitats of *Osmanthus delavayi* expand to the west and north, and the highly suitable habitats decrease. Zhang et al. (2018) [38] found that the overall geographical distribution area of *Sorbus amabilis* contracted, and the degree of fragmentation increased and migrated to high-altitude areas in the future. The results of this study are basically consistent with the above conclusions. The suitable habitats of *M. wufengensis* in the future showed characteristics of habitat fragmentation, and the centroids of the suitable habitats generally moved to high elevations in the northeast direction. The main limitation of the potential habitat distribution of *M. wufengensis* is the low-temperature factor. In the future, the temperature in northern China will rise, and *M. wufengensis* will be able to survive winter in some areas, so the potential suitable habitats will expand to high-latitude areas. In Guizhou, Yunnan, Guangxi, Sichuan, Hubei, Hunan, Anhui, Taiwan, and other places with high mountain ranges, the areas of suitable habitat significantly increased. The mountain ranges in these areas have a high average elevation, and the temperature will increase due to global warming in the future, which can meet the growth conditions of *M. wufengensis*. Therefore, the potential suitable habitats for *M. wufengensis* will migrate to higher altitudes at the same time.

In summary, judicious utilization of SDMs can effectively and swiftly discern the potential distribution areas of rare and endangered species, facilitating the formulation of subsequent conservation policies. Additionally, these models can anticipate the species' future distribution based on environmental factors in forthcoming time frames, enabling the proactive development of corresponding conservation plans and thereby maximizing species richness preservation. However, in practical applications, certain models rely solely on algorithmic logic, lacking robust ecological interpretations [74]. Some models even depend solely on expert experience, exhibiting considerable subjectivity [33]. Moreover,



the influence of sample quality and quantity is significant for some models [33]. There-fore, during the actual modeling process, the selection of the most suitable model should be based on the algorithms and theoretical underpinnings specific to different species distribution models.

## 5. Conclusions

In this study, four SDMs (Maxent, GARP, Bioclim, and Domain) were used for the first time to comprehensively predict and analyze potential suitable habitats for the introduction of the rare and endangered plant *M. wufengensis* in China. The GARP model simulated the widest range of suitable habitats. The Domain model simulated the smallest range, and its layers were not clear. The Bioclim model results were similar to those of the Maxent model, and the Maxent model had the best performance and the best simulation effect based on the AUC and Kappa statistics. The low-temperature factor was the dominant environmental factor affecting the distribution of *M. wufengensis* in China. The potential suitable habitat for *M. wufengensis* was mainly distributed in the areas south of 40° N and east of 97° E in China, with a high distribution potential under current climate conditions. Under future climate scenarios, the highly suitable habitats for *M. wufengensis* in China generally showed a decreasing trend, and moderately and lowly suitable habitats showed an increasing trend. The centroid of the future potential suitable habitats of *M. wufengensis* migrated to the northeast at a high latitude.

**Supplementary Materials:** The following supporting information can be downloaded at https://www.mdpi.com/article/10.3390/f14091767/s1. Table S1: Environmental variables used in this study; Table S2: The performance of Maxent model using default settings and optimized settings by EN-Meval. Table S3: The contribution and permutation importance of 16 environmental factors.

**Author Contributions:** Investigation, X.S. and Z.Z.; methodology, X.S. and Q.Y.; formal analysis, X.S. and Z.Z.; data curation, Z.S.; writing—original draft preparation, X.S. and Q.Y.; writing—review and editing, Z.S., Z.Z., Z.J. and L.M.; supervision, Z.S. and L.M.; project administration, Z.J. and L.M.; funding acquisition, X.S., Z.J. and L.M. All authors have read and agreed to the published version of the manuscript.

**Funding:** This research is supported by the Zhejiang Provincial Scientific Research Institute Special Project (2023F1068-4), the Special Fund for Forest Scientific Research in the Public Welfare under grant no. 201504704, and the Transformation and application of forestry intellectual property project under grant no. Intellectual property transformation 2017-11.

**Acknowledgments:** We are thankful to Wufeng Bo Ling *Magnolia wufengensis* Technology Development Co., Ltd. for providing help in the field survey.

**Conflicts of Interest:** The authors declare no conflict of interest.

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
