# Peer review of "Habitat Distribution Pattern of Rare and Endangered Plant Magnolia wufengensis in China under Climate Change"

_forests, doi:10.3390/f14091767_

Round 1

Reviewer 1 Report

Comments and Suggestions for Authors

I found this manuscript very interesting and well-written. The quality and number of graphs are adequate for this type of paper. I have only  one suggestion: In L 124 & 125 the word "ex-situ" should be written as "ex situ", that is, in italics and without de - separating both words.

Author Response

Response: We revised "ex-situ" into "ex situ" in the new line 124 & 125.

Reviewer 2 Report

Comments and Suggestions for Authors

The article is well defined in all its parts.

It has a sufficiently broad introduction and good citations.

The methodology is well exposed, giving way to the results following the same methodology.

In the results section a table with potential surface area values could be added.  In this table a comparison can be made between the current surface (with the 4 levels) and the future surfaces, which would allow to see the increase and/or decrease of the potential surface in the different scenarios.

I think that in this way, the results can be seen in a clearer way.

In general the article is fine, and if you want to take into account what was said above it would be ready to be published.

Reviewer 3 Report

Comments and Suggestions for Authors

Round 2

Reviewer 3 Report

Comments and Suggestions for Authors

The manuscript is sufficiently improved.